# Intracranial Aneurysms and Genetics: An Extensive Overview of Genomic Variations, Underlying Molecular Dynamics, Inflammatory Indicators, and Forward-Looking Insights

**DOI:** 10.3390/brainsci13101454

**Published:** 2023-10-12

**Authors:** Corneliu Toader, Lucian Eva, Bogdan-Gabriel Bratu, Razvan-Adrian Covache-Busuioc, Horia Petre Costin, David-Ioan Dumitrascu, Luca-Andrei Glavan, Antonio Daniel Corlatescu, Alexandru Vlad Ciurea

**Affiliations:** 1Department of Neurosurgery, “Carol Davila” University of Medicine and Pharmacy, 020021 Bucharest, Romania; corneliu.toader@umfcd.ro (C.T.); razvan-adrian.covache-busuioc0720@stud.umfcd.ro (R.-A.C.-B.); horia-petre.costin0720@stud.umfcd.ro (H.P.C.); david-ioan.dumitrascu0720@stud.umfcd.ro (D.-I.D.); luca-andrei.glavan0720@stud.umfcd.ro (L.-A.G.); antonio.corlatescu0920@stud.umfcd.ro (A.D.C.); prof.avciurea@gmail.com (A.V.C.); 2Department of Vascular Neurosurgery, National Institute of Neurology and Neurovascular Diseases, 077160 Bucharest, Romania; 3Department of Neurosurgery, Dunarea de Jos University, 800010 Galati, Romania; 4Department of Neurosurgery, Clinical Emergency Hospital “Prof. Dr. Nicolae Oblu”, 700309 Iasi, Romania; 5Neurosurgery Department, Sanador Clinical Hospital, 010991 Bucharest, Romania

**Keywords:** intracranial aneurysm, genetic syndromes, genome-wide association studies, molecular mechanisms, inflammatory status, genomic modifications

## Abstract

This review initiates by outlining the clinical relevance of IA, underlining the pressing need to comprehend its foundational elements. We delve into the assorted risk factors tied to IA, spotlighting both environmental and genetic influences. Additionally, we illuminate distinct genetic syndromes linked to a pronounced prevalence of intracranial aneurysms, underscoring the pivotal nature of genetics in this ailment’s susceptibility. A detailed scrutiny of genome-wide association studies allows us to identify key genomic changes and locations associated with IA risk. We further detail the molecular and physiopathological dynamics instrumental in IA’s evolution and escalation, with a focus on inflammation’s role in affecting the vascular landscape. Wrapping up, we offer a glimpse into upcoming research directions and the promising horizons of personalized therapeutic strategies in IA intervention, emphasizing the central role of genetic insights. This thorough review solidifies genetics’ cardinal role in IA, positioning it as a cornerstone resource for professionals in the realms of neurology and genomics.

## 1. Introduction

About 80% of all non-traumatic subarachnoid hemorrhages (SAHs) are caused by intracranial aneurysms (IAs), commonly referred to as saccular or berry aneurysms. Between 2% and 5% of people have intracranial aneurysms. Of these, 0.7% to 1.9% will experience a rupture leading to subarachnoid hemorrhage [1,2].

An SAH is the predominant initial sign of cerebral aneurysms in both adults and kids. In children, the incidence of an SAH varies between 1.9% and 4.6%. The growing detection of SAHs in children can probably be attributed to better diagnostic tools and heightened clinical vigilance [3].

However, the occurrence of an IA seems to increase with age [4]. For those over 30 years old, the likelihood of having an IA ranges from 3.6% to 6.5% [5]. Women are more predisposed to aneurysms than men, with a 3:1 ratio in cases of unruptured IAs. About 70–75% of IAs appear individually, whereas 25–30% occur as multiple aneurysms [6,7].

Among adults, unruptured IAs have a prevalence rate of about 2% to 6%. They generally do not show symptoms and are usually discovered during MRI or CT scans for other neurological reasons or when screening high-risk individuals [8].

The frequency of SAHs due to IAs varies globally. Finland and Japan have the highest rates, between 22.5 and 32 per 100,000 people annually. Globally, this rate stands at 9.1 per 100,000 individuals every year [9]. An SAH strikes early in life and is fatal, contributing to more than 25% of years of life lost for stroke sufferers below 65 years of age [10].

The majority of individuals are unaware that they have an aneurysm until it bursts. Some might experience precursor symptoms like pain around the eye, nerve paralysis in the face, or headaches and neck pain from a minor aneurysm leak, known as a “sentinel bleed” [11].

## 2. Risk Factors Associated with IAs

Habits and health conditions such as active smoking, a high blood pressure, and excessive alcohol consumption are recognized as separate risk factors that can contribute to the development of an aneurysmal subarachnoid hemorrhage [12]. Additional risk factors include an advancing age and being female, since the yearly occurrence rate for men was determined to be 4.5 per 100,000 with a confidence interval (CI) of 3.1 to 5.8, while for women, it was 7.1 per 100,000 with a CI of 5.4 to 8.7. The risk for women, when compared to men, had a relative value of 1.6 with a CI of 1.1 to 2.3 [13]. Moreover, having a family history of intracranial aneurysms, and the consumption of drugs that stimulate the sympathetic nervous system, commonly known as sympathomimetic drugs, are other risk factors [14]. Genetic conditions can further heighten the risk of an aneurysmal SAH. For instance, those with autosomal dominant polycystic kidney disease or vascular Ehlers-Danlos syndrome (often referred to as type IV EDS) are more susceptible.

Recent research, like the PHASES study, indicates that one’s geographical origin, such as being of Finnish or Japanese descent, can be a strong determinant for aneurysmal rupture. This potentially suggests that genetic predispositions, associated with certain regions or ethnicities, can play a significant role in rupture risk [15]. The 1999 study by the MARS group provided clarity on the importance of screening close relatives of a patient diagnosed with an IA. Their findings underscored the preventive value of screening: for every 149 first-degree relatives screened using Magnetic Resonance Angiography (MRA), one SAH could be averted. To prevent a single SAH-related death, 298 patients would need to be screened [16]. A family history of IA stands out as the most telling risk marker for the condition. To illustrate, family members from a family with at least two patients diagnosed with IA have about a four-fold increased risk of having an IA, as shown by MRA screening, compared to the general public [4].

Although several inheritable conditions, including autosomal dominant polycystic kidney disease, neurofibromatosis type I, Marfan syndrome, multiple endocrine neoplasia type I, pseudoxanthoma elasticum, hereditary hemorrhagic telangiectasia, and Ehlers-Danlos syndrome types II and IV, have links to the formation of IAs, they represent less than 1% of all IAs in the general population. As such, these genetic syndromes cannot account for the majority of familial clusters of IA cases. However, when family histories are thoroughly researched, familial occurrences of IA seem to be more frequent than initially thought. By studying extended family trees over multiple generations based on affected individuals, we might quickly identify a significant number of familial cases [17].

### 2.1. Genetic Syndromes with Increased Intracranial Aneurysm Incidence

*Autosomal Dominant Polycystic Kidney Disease (ADPKD)*: Through comprehensive literary reviews, it has been established that individuals with ADPKD have a roughly 11% prevalence rate of intracranial aneurysms [18]. ADPKD’s genetic origin lies in the loss-of-function of either the polycystin 1 (PKD1) or polycystin 2 (PKD2) genes. This primarily results in the development of cysts in the kidneys, which can impair renal function and eventually progress to kidney failure [19]. Beyond the kidneys, ADPKD can impact other organs, particularly the liver. A common comorbidity among ADPKD patients is hypertension, which is a known risk factor not just for IA, but for other cardiovascular diseases as well [20]. The life expectancy for those with ADPKD is notably reduced, with PKD1 and PKD2 patients in Europe averaging lifespans of 53 and 69 years, respectively [21].

*Microcephalic Osteodysplastic Primordial Dwarfism Type II (MOPDII)*: This is the most prevalent variant of microcephalic primordial dwarfism. It presents with an exceptionally short stature, microcephaly, and unique facial attributes. The features that distinguish it from other primordial dwarfism types, which might require medical attention, encompass irregular dental structures, a fragile bone skeletal dysplasia leading to hip deformities or scoliosis, tendencies towards insulin resistance or diabetes, chronic kidney ailments, heart defects, and a broad spectrum of vascular diseases. This wide-ranging vascular ailment category encompasses neurovascular conditions like moyamoya vasculopathy and intracranial aneurysms (potentially causing strokes), early-onset coronary artery diseases (potentially resulting in premature heart attacks), and kidney-related vascular issues. The frequent occurrence of hypertension is complicated due to multiple potential origins tied to the array of associated conditions [22].

Observationally, from birth onwards, 25 out of 47 individuals were identified with intracranial aneurysms (30 of them had multiple aneurysms), 22 showed signs of moyamoya vasculopathy, 17 exhibited both conditions, and 17 had neither. Roughly 50% of those diagnosed with either moyamoya vasculopathy or aneurysms eventually experienced a stroke, be it ischemic or hemorrhagic. The risk of an aneurysm remained fairly consistent throughout childhood, while the risk associated with moyamoya vasculopathy was accentuated before the age of five [23].

*Type IV Ehlers–Danlos Syndrome (Vascular Subtype)*: This autosomal dominant condition, affecting between 1 in 50,000 to 200,000 individuals, is a connective tissue disorder. It is chiefly marked by pronounced vascular fragility, heightening the risk of a hemorrhage and premature death [24]. The underlying cause of vascular EDS is the presence of mutations or variants in the Col3a1 gene. This gene is responsible for producing a protein that plays a crucial role in reinforcing connective tissue throughout the body [25].

Microcephalic/Majewski’s Osteodysplastic Primordial Dwarfism, Type II (MOPD2): This exceedingly rare autosomal recessive disorder is characterized by a notably short stature, alongside skeletal abnormalities, especially a significantly small head size in proportion to the body [26]. Given the scant occurrence of MOPD2, establishing a precise prevalence of IA in affected individuals remains challenging. However, current evidence indicates that as many as half of those with MOPD2 might develop an IA [27]. The genetic basis for MOPD2 is traced back to mutations in the Pericentrin 1 (PCNT) gene. This gene encodes for a protein that is central to chromosomal segregation. Additionally, a loss-of-function in PCNT disrupts cilia formation in epithelial cells and impacts PDK2 positioning at the basal bodies [28]. The intricate interaction between PCNT and PDK2 and the potential implications of PCNT insufficiency in an IA becomes even more evident when considering the presence of rare PCNT variants in multiple IA-affected families. Some family members also presented with kidney cysts [29].

However, in these families, it is crucial to understand that other genetic variants, apart from PCNT, may exist. These additional variants might play significant roles in disease manifestation and progression.

*Loeys–Dietz Syndrome (LDS)*: LDS shares similarities with type IV Ehlers–Danlos, being an autosomal dominant connective tissue disease. Caused by mutations in the genes of the Tgf-β pathway (mainly TGFBR1, TGFBR2, SMAD3, and TGFB2), patients exhibit severe vascular anomalies. Arterial aneurysm dissections and bleeding events, which often appear early in life, are frequent complications. A significant portion, roughly one-third, of LDS-related deaths, result from cerebrovascular hemorrhages. The incidence of an IA among those with LDS ranges from 10% to 28% [30].

*Marfan Syndrome* is among the prevalent hereditary disorders impacting connective tissue. It is an autosomal dominant condition, manifesting in approximately 1 out of every 3000 to 5000 individuals. The underlying genetic flaw is in the FBN1 gene on chromosome 15, which is responsible for producing fibrillin, a vital connective tissue protein. The heightened risk associated with the aorta in males has been recognized for a while, yet its underlying reason remains unclear. In this context, the influence of gender was evident across all molecular categories. Patients with PTC displayed a trend towards earlier surgeries, and the lifetime aortic risk for males was double compared to females who possessed in-frame pathogenic variations [31]. Though some studies suggest a link between Marfan’s syndrome and IAs, autopsies and family analyses with multiple members having both an IA and Marfan’s syndrome do not consistently support this association [32,33].

*Neurofibromatosis type 1*: NF1 is a condition marked by tumors in the skin and nervous system. The two primary forms, types 1 and 2, are both inherited in an autosomal dominant manner. Type 1, often referred to as von Recklinghausen disease, is typified by features like neurofibromas, cafe-au-lait spots, freckling, and optic gliomas. In contrast, type 2 is distinguished by the presence of bilateral vestibular schwannomas and meningiomas [34].

In a 2005 research study by Schievink et al. [35], 39 patients with neurofibromatosis type 1 were examined, with an average age of 30.4 years, ranging from 3 to 64 years. Incidental intracranial aneurysms were found in 2 (5%) out of the 39 patients through MRI scans. When focusing only on the 22 patients who had an MRI, the detection rate rose to 9%. This rate was notably higher (*p* < 0.005) than the 0% detection rate in the control group of 526 individuals, therefore concluding the existence of a correlation between NF1 as a genetic risk factor for IA development.

*CDKN2BAS*: The 9p21.3 locus first gained attention during a 2007 genome-wide association study (GWAS) related to cardiovascular disease. In this study, it was highlighted that the specific genetic variations associated with cardiovascular diseases also had a connection with intracranial aneurysms (IAs). Interestingly, the 9p21.3 region itself does not house any known protein-coding genes. However, in its vicinity—just a few kilobases away—are protein-coding genes, including CDKN2A and CDKN2B [36]. It is worth mentioning that the 9p21.3 gene cluster (CDKN2A-CDKN2B) has emerged as a frequently identified region in GWASs for several prevalent conditions, such as heart diseases, type 2 diabetes, brain tumors (gliomas), and skin cancers (basal cell carcinomas) [37].

One significant discovery came from a study that pinpointed the CDKN2BAS SNP (rs6475606) as a contributing factor for IA vulnerability, particularly in the US population. This was a monumental finding, as understanding genetic predispositions can be the key to predicting and potentially preventing the onset of conditions like IAs [38].

Subsequent GWAS research involving both Asian and Caucasian populations has further underscored this genetic link. Numerous single-nucleotide polymorphisms (SNPs) of CDKN2BAS have been identified as being associated with intracranial aneurysms. This consistent finding across diverse populations suggests a robust genetic connection and underscores the importance of this genetic region in understanding, and potentially addressing, IAs [39].

*Other Syndromes*: Several conditions, such as multiple endocrine neoplasia type I [40], pseudoxanthoma elasticum [41], and hereditary hemorrhagic telangiectasia (HHT) [42], are frequently highlighted in relation to intracranial berry aneurysm. While there are individual patient case studies documenting the existence of intracranial aneurysms in these conditions, the evidence is not as comprehensive as it is for other syndromes that were previously discussed [43].

### 2.2. Genetic Implications

A predominant strategy for discerning genetic risk factors related to intracranial aneurysms (IAs) entails genotyping polymorphisms within sporadic cases and controls. The genetic markers frequently employed in these endeavors are termed single nucleotide polymorphisms (SNPs). Genetic association investigations can be segmented into two main categories: Candidate Gene Association Studies (CGASs) and genome-wide association studies (GWASs). A CGAS involves the examination of select common polymorphisms in genes, specifically chosen based on existing biological evidence suggesting their relevance to IA development. Additionally, candidate genes for IAs have been sourced from genetic studies focusing on connective tissue disorders like vascular Ehlers–Danlos syndrome; known genetic diseases where an IA is a manifestation, such as polycystic kidney disease; and from gene expression studies [44].

A plethora of recognized mendelian disorders is known to elevate the risk of IA development. Over recent decades, there has been a recalibration in our understanding of genetic causation. Previously, the paradigm centered on single gene mutations that are directly correlated with distinct clinical phenotypes. However, the contemporary perspective emphasizes the intricate interactions between singular gene mendelian phenotypes and allelic variants located elsewhere in the genome, termed transacting regulatory elements. Consequently, our understanding of genetic inheritance in diseases has evolved from a binary approach to a more nuanced, spectrum-oriented view [45].

The proclivity towards IAs, stemming from these genetic anomalies, likely emanates from structural deficiencies in the brain’s vasculature due to the malfunctioning of specific genes. In illuminating this, a recent meta-analysis concerning the ACE insertion/deletion (I/D) polymorphism (rs4646994) demonstrated a linkage between the I allele and increased IA susceptibility. Specifically, individuals possessing ACE I/I and I/D genetic configurations displayed a heightened IA risk. The exact mechanisms by which the ACE I allele influences the development of an IA remain nebulous [46].

Enriching the discourse, Yasuno and colleagues amalgamated more datasets, thereby analyzing a larger sample consisting of 5891 cases and 14,181 controls. Their findings revealed a total of five IA-specific loci. Among these, the SOX17 (8q11.23; rs9298506) and CDKN2BAS1 (9p21; rs1333040) loci were previously identified, and their association with IAs was fortified in this research. Although the SNP rs10958409 (8q11.23; rs10958409) did not manifest an association, the study unveiled three novel IA associations with CNNM2 (10q24.32; rs12413409), STARD13 (13q13.1; rs9315204), and RBBP8 (18q11.2; rs11661542). It is pivotal to acknowledge that certain SNPs, despite achieving genome-wide significance like rs700651 and rs11661542, did not exhibit replicability in subsequent studies and thus were not deemed significant in the encompassing meta-analyses [47] (Figure 1 and Table 1).

There exist certain genetic syndromes that are transmitted through an autosomal recessive manner or stem from spontaneous genetic alterations. These are known to be associated with intracranial aneurysms, albeit at a lesser frequency compared to more prevalent syndromes. An exemplar of such conditions is pseudoxanthoma elasticum. This disorder is linked to mutations found within the ABCC6 gene situated on chromosome 16p13 [48]. One prevailing theory proposes that IAs in individuals with this syndrome originate due to irregularities in the elastic lamina of the intracranial blood vessels, with the intracranial internal carotid artery being a primary site [41]. Whole exome sequencing (WES) has spotlighted the potential significance of the PCNT gene in relation to cerebrovascular diseases. The protein encoded by this gene plays a pivotal role in orchestrating microtubule nucleation and ensuring the orderly progression of the cell cycle. Importantly, mutations where both copies of the gene are dysfunctional lead to a condition known as microcephalic osteodysplastic primordial dwarfism type II. Astonishingly, in over half of these cases, it is surmised that individuals bearing specific mutations in the PCNT gene are susceptible to IAs. Intriguingly, these genetic variations might also amplify the risk of a single individual manifesting multiple IAs [27].

While these genetic anomalies offer invaluable insights into the genesis of aneurysms, comprehending the specific risks tied to aneurysm rupture in these distinct syndromes necessitates further exploration. Beyond merely identifying the genetic markers related to the onset or presence of IAs, there exists a trove of data ripe for analysis to pinpoint genetic locations that might dictate the propensity for aneurysmal rupture and the subsequent clinical aftermath of a subarachnoid hemorrhage (SAH). The post-rupture genetic landscape of the disease becomes murkier, as there could be other genetic factors influencing various aspects of SAH care. One such aspect is vasospasm—a contraction of blood vessels—which is widely acknowledged as a key contributor to grievous outcomes due to delayed cerebral ischemia (DCI) [49]. Presently, there is a noticeable scarcity of comprehensive studies elucidating outcomes post aneurysm rupture. The limited studies available in the scientific literature often lack the rigorous sample sizes required for drawing incontrovertible conclusions. Most of these outcome-centric studies place their emphasis on the ever-fluctuating gene expression profiles. This dynamic nature contrasts with static genetic anomalies, which could potentially offer a constant measure of risk, hence posing challenges in consistently assessing the genetic basis of the outcomes [50].

### 2.3. Genome-Wide Association Studies

Genome-wide linkage analyses are sophisticated methods utilized to pinpoint the specific chromosomal location, referred to as loci, that houses the gene that is responsible for certain diseases. To accomplish this, these analyses probe specific genetic markers, such as single nucleotide polymorphism (SNP) and Variable Number Tandem Repeat (VNTR) [51]. Thanks to the versatility of the linkage analysis method, researchers have managed to unveil genes that drive the onset of complex disorders. These range from metabolic conditions like diabetes and obesity to cardiovascular issues such as hypertension. The breakthroughs provided by these analyses have deepened our understanding of the genetic underpinnings of such multifaceted ailments [52,53]. Over time, numerous studies have leveraged linkage analysis to unearth insights about IAs. The allure of non-parametric methodologies is particularly potent when it comes to IAs. This is primarily because the penetrance—referring to the likelihood that an individual with a mutation will display symptoms—may not be absolute. In simpler terms, not every individual harboring the mutation will be afflicted. Summarily, non-parametric linkage studies have unveiled several loci that may play roles in the formation and potential rupture of IAs. However, only a select few, namely 1p34.3-p36.13, 7q11, 19q13.3, and Xp22, have seen consistent results across various populations [54,55,56].

Current research furnishes the most compelling evidence concerning regions on chromosomal arms 7q and 19q. Numerous independent studies have reported linkage hits on these arms, signaling their potential significance in IAs. Alg et al., in a comprehensive meta-analysis involving a staggering 116,000 participants, spotlighted 19 SNPs that exhibited a significant correlation with IAs under at least one genetic model. GWASs emerged as particularly illuminating, uncovering 12 robustly associated SNPs. Some of the most telling associations are related to chromosomes 9, 8, and 4. Meanwhile, the CGAS method identified eight significant SNPs, with SERPINA3 and two collagen-related variants showcasing the strongest links. Notably, 9 out of the 19 SNPs exhibited significant statistical variance, mandating random-effects and sensitivity evaluations for those reported in over two publications [57].

A recent genome-wide association study, centered on both familial and sporadic IAs in a Caucasian population, identified six SNPs in the 9p21.3 region as being significantly associated with IA. Of these, one (rs6475606) achieved a stellar level of statistical significance for its association with both types of IAs. Additionally, the study reaffirmed the association of the rs1333040 SNP with IAs, bolstering the belief in its relevance [58].

The endothelin receptor type A (often abbreviated as EDNRA) gene encodes a receptor that is activated by endothelins, a group of proteins instrumental in regulating both the constriction and dilation of blood vessels, particularly following any hemodynamic disturbances. Specifically, Endothelin-1, which is the primary variant present in vascular smooth muscle cells, is responsible for triggering EDNRA. This endothelin signaling pathway becomes particularly active at sites of vascular injuries, leading to heightened cell proliferation—a critical step in the body’s reparative process [59]. There is a potential that a diminished activity, or downregulation, of EDNRA signaling might be a precursor to compromised vascular repair mechanisms. When this repair process does not function optimally, it could make the vascular system more susceptible to the formation of aneurysms following injuries or disturbances. Adding credence to this theory is the functional analysis of the EDNRA gene variants. Specifically, the rs6841581 risk allele has been shown to have reduced transcriptional activity. This essentially means that the gene’s ability to produce its associated protein might be diminished, which could potentially contribute to the aforementioned vulnerabilities in the vascular repair system [60].

Continuing research using genome-wide association studies recently highlighted another intriguing discovery. The SNP denoted as rs6841581A.G, located on chromosome 4q31.23, which codes for the EDNRA gene, has caught researchers’ attention. This particular SNP displayed a significant association with intracranial aneurysms. What is particularly noteworthy is that this association was not restricted to just one ethnic group or population; it emerged as a consistent pattern across Dutch, Finnish, and Japanese populations. This widespread correlation accentuates the potential global relevance and importance of the EDNRA gene in understanding and potentially addressing IAs [61] (Table 2).

## 3. Molecular and Physiopathological Mechanisms Implicated in IA Formation and Progression

Human IA samples serve as a treasure trove for researchers, offering them a unique glimpse into the intricate molecular mechanisms underpinning an IA’s onset and rupture. At the heart of IA pathology is the degradation of the internal elastic lamina (IEL), which sits as a protective barrier separating the intima from the media layers of blood vessels. Alongside this, various other signs, like an uneven vessel inner surface, thickening due to myointimal growth, the chaotic nature of the muscular media, reduced cell presence, and heightened inflammatory cell infiltration, paint a complex picture of an IA’s internal environment [62].

As a critical component in arteries, the IEL comprises elastic fibers that grant flexibility to these vessels. In a healthy state, the IEL presents as consistent and intact in intracranial arteries. However, when confronted with an IA, the IEL tends to become fragmented or torn, or may even vanish completely, which is especially evident at the aneurysm’s base. This degradation is a significant red flag pointing towards an IA’s pathology [63].

One of the more striking aspects of IAs, whether from human patients or those mimicked in animal models, is the formation of pronounced outward bulges and deep inward crevices in the intimal lining. These anomalies become evident through the intricate imaging of transmission electron microscopy [64].

The degradation of the IEL acts as a trigger. It allows for the migration of smooth muscle cells from the media to the intima layer. Here, they multiply, leading to myointimal hyperplasia, resulting in a pronounced thickening of the intima layer, which is a recurring characteristic in IA samples [65].

IAs mark a shift in the medial layer’s cellular configuration. Here, smooth muscle cells transition or “switch” from their primary contractile state to a more synthetic one. This new state is marked by heightened inflammatory and remodeling tendencies. In the context of an IA, there is a heightened influx of inflammatory cells. Both animal models and human samples consistently reveal the presence of cells like macrophages, T cells, B cells, and neutrophils. Significantly, macrophages, apart from their standard functions, release matrix-degrading enzymes such as MMP2 and MMP9. These enzymes, alongside various cytokines, serve to attract a greater volume of inflammatory cells to the site [64].

Microarray-based mRNA profiling stands out as a holistic tool, allowing for the clear demarcation of molecular markers in both healthy and diseased states. While arteries have their distinct mRNA expression profiles, the same type of artery from varied individuals tends to have more similarities than different arteries within one individual. This highlights the necessity for meticulous control tissue selection when conducting expression studies [66].

To truly discern the molecular markers specific to an intracranial artery, researchers utilized RNA samples from both IA-affected and unaffected arteries, applying two microarray platforms. Intriguingly, almost half of the genes on these platforms were expressed in these arteries. Further, DNA linkage studies spotlighted around 800 diverse genes present in intracranial arteries. Utilizing advanced bioinformatics tools like GO and KEGG, researchers found a significant clustering around biological pathways, including the likes of Notch and MAPK signaling channels [67].

Genome-wide studies focusing on both mRNA and miRNA expressions offer a comprehensive, unbiased pathway to dive into an IA’s pathophysiology. Tools like GO and KEGG not only facilitate a more structured understanding of gene expression, but also throw light on the intricate web of interconnected pathways, providing a holistic view of the underlying pathology defining an IA.

## 4. Adherens Junction, MAPK, and Notch Signaling, Which Are Functionally Relevant to the Pathogenesis of an IA

**Adherens Junction:** When delving into the intricate world of intracranial aneurysms, the most pivotal biological pathway that emerges is the adherens junction pathway. These adherens junctions, far from being static structures, are vibrant, multifaceted assemblies comprising both adhesive and signaling molecules. Their primary roles are to oversee and maintain the inhibition of endothelial cell growth in response to contact. Additionally, they play pivotal roles in regulating vascular permeability, ensuring a controlled environment against inflammatory cells and solutes. Another feather in their cap is their responsibility in steering the formation of new blood vessels during angiogenesis. Zooming in on vascular endothelial cells, adherens junctions wear the badge of honor in preserving the vessel wall’s sanctity. They also play crucial roles in remodeling the endothelium during both physiological and pathological events. An interesting tangent to this is how proteins within these junctions can channel signals to β-catenin, which can then make its journey to the nucleus. Once there, it synergizes with transcription factors to govern the expression of genes that are instrumental in the endothelial cell’s life cycle, influencing their growth, differentiation, and programmed cell death [68]. What catches one’s eye is the CTNND2 gene. Located on chromosome 5p and aligned with the IA locus, it encodes for catenin delta-2. This protein could potentially be the puppet master in regulating adhesion molecules within the brain’s confines. Drawing a parallel, another gene, CTNNA1, which encodes catenin alpha-1, is also seen as a prominent positional IA candidate. This particular gene might be tasked with ensuring that the integrity of the intracranial arteries remains uncompromised [69].

In certain IA patients, a rather unique phenomenon can be observed: the presence of red blood cells not just in the intima but also the media. One could theorize that such dissections could stem from underlying defects in the cohesion and integrity between endothelial cells, involving the trifecta of adherens, tight, and gap junctions. Offering further insight, mouse models have demonstrated that the absence of SOX17 in endothelial cells paves the way for IA formation. Furthermore, it disrupts VE-cadherin, a pivotal element in adherens junctions. This gives rise to the notion that proper cell-to-cell adhesion might be a protective barrier against an IA’s formation. Diving deeper, it is believed that SOX17, acting as a transcription factor, might have a diverse set of downstream targets, with VE-cadherin and its regulators being among them [70].

Another intriguing gene that emerges in the context of IAs is THSD1. This gene holds the reigns to the assembly of adherens junctions, as evidenced by reduced or misplaced VE-cadherin in endothelial cells devoid of THSD1. As researchers peer into the future, a tantalizing prospect is uncovering whether THSD1 also operates under the influence of SOX17. This could potentially unveil an additional layer of governance over VE-cadherin’s activity within endothelial cells [71].

Taking a deeper dive into THSD1 reveals it as a single-span transmembrane protein that intricately intertwines with FAK signaling and cellular adhesion. Proteomic studies originally spotlighted Thsd1 as a potential partner of talin based on mass spectrometry analyses of talin immunoprecipitation [72].

Current databases detailing protein–protein interactions have not yet painted a complete picture concerning Thsd1 interactions. A significant reason behind this gap might be the predominant presence of THSD1 in endothelial cells, thereby limiting the breadth of available data. But given the monumental role that talin plays in integrin-mediated focal adhesion, it is plausible that THSD1 could influence endothelial focal adhesion and FAK signaling. To further bolster this hypothesis, experiments on human umbilical vein endothelial cells have pointed towards THSD1′s regulatory role over focal adhesion stability. When THSD1 was silenced using RNA interference techniques, there was a noticeable dip in the number of focal adhesions, paralleled by a reduced capability of cells to adhere to collagen. Interestingly, subsequent rescue experiments threw light on how most THSD1 variants exhibited a diminished ability in both focal adhesion and cell attachment [62] (Figure 2).

**MAPK Signaling**: The MAPK (Mitogen-Activated Protein Kinase) pathway is a central hub of biological signaling, with a vast interconnected network. This intricate pathway splits into three main branches: the classical MAPK pathway, the c-Jun NH2-terminal kinase (JNK) and p38 MAPK pathway, and the ERK5 pathway. All of these play crucial roles in managing a cell’s life cycle, covering everything from its proliferation and differentiation to inflammation and programmed cell death. When it comes to the cardiovascular system, MAPKs are present in various vascular cell types such as cardiomyocytes, vascular endothelial cells, and vascular smooth muscle cells (VSMCs). They are entrusted with the pivotal task of controlling cardiovascular signal transduction pathways. These kinases act as translators, converting signals from cytokines, growth factors, and environmental stressors (like ischemia and shear stress) into cellular responses. In-depth studies of the cardiovascular system have underscored the significance of these MAPK cascades, which govern various vital functions, including regulating vascular endothelial cell permeability, producing cytokines, modulating vasomotor function, and mediating the effects of reperfusion injuries [73]. In the context of intracranial aneurysms, 18 identified candidate genes have been found that are intricately linked with MAPK signaling. Some of the key players include MAPK7, TGFB3, and CDC42. Interestingly, genes encoding heat shock proteins, such as HSPA9B and HSPB1, also make their presence felt in IA. Further widening the MAPK network in intracranial arteries, other significant genes include EGFR, TGFBR1, and IL1B. Within this intricate web, HSP5A emerges as a potential candidate gene for IAs, with a major role in negatively regulating JNK. Similarly, DUSP10 encodes a dual-specificity MAPK phosphatase, functioning primarily in both innate and adaptive immune responses by inhibiting JNK and thus influencing the transcription factor activator protein-1 (AP1) [67].

**Notch Signaling**: The Notch signaling pathway wears multiple hats. It is instrumental in the development of cardiovascular structures within mammals, facilitating the formation of arteries and veins by acting upon endothelial and smooth muscle cells. This pathway’s significance does not wane in adult vascular systems. Mutations in the NOTCH3 gene have been associated with cerebral autosomal dominant arteriopathy with subcortical infarcts and leukoencephalopathy (CADASIL). This condition stands out as the predominant hereditary stroke disorder [74]. Underlying CADASIL is the progressive degeneration of VSMCs within arteries. Mutations within the NOTCH3 gene lead to the anomalous accumulation of the Notch3 receptor at the VSMCs’ cytoplasmic membrane within intracranial vessels [75]. CADASIL patients exhibit disruptions in the typical anchoring of VSMCs to the extracellular matrix and neighboring cells. Coupled with alterations to the cytoskeleton, this might set off the chain reaction, leading to VSMC degeneration [76]. VSMC loss brings about abnormalities in the endothelial cell tight and gap junctions. Such adhesion pathways are interwoven with others, and are responsible for maintaining cell junctions, involving not just adherens junctions but also cell adhesion molecules, actin cytoskeleton regulation, and the previously mentioned MAPK and Ca2 signaling pathways [77].

An intracranial aneurysm exhibits marked differences in the collagen gene expression levels, with a spike in transcription for TIMP-3. These observations hint at extensive extracellular matrix (ECM) remodeling within the aneurysm, aligning with the findings of elevated levels of matrix metalloproteases (MMPs)-2 and -9 in aneurysmal tissues [78].

Intracranial aneurysms showcase an overexpression of the factors believed to be active participants in collagen metabolism, including big-h3 and CTGF. The exact role of big-h3 remains an enigma, though it is thought to potentially bridge collagen interactions within the ECM [79]. Though big-h3′s exact function remains a topic of ongoing research, it is postulated that it might serve as a bridge, facilitating or stabilizing the interactions between collagen and other integral ECM structures [80].

Another molecule of interest in this context is SPARC (osteonectin), a counter adhesive glycoprotein found across various tissues, from vascular endothelium and smooth muscles to fibroblasts. Beyond inhibiting endothelial cell adhesion and proliferation, elevated SPARC protein and mRNA levels have been documented in renal vascular injury scenarios [81]. The intricate dance of ECM remodeling may be choreographed, in part, by specific interactions between SPARC and type I collagen, highlighting the delicate balance and interplay within the vascular system [82]. Intriguingly, within the realm of acidic proteins, another significant member is hevin. Located primarily in the high endothelial venules of lymphoid tissues, hevin possesses antiadhesive properties. What is captivating is its striking similarity to SPARC. Drawing a parallel with SPARC, hevin’s expression is noticeable within intracranial aneurysms, yet it remains conspicuously absent in STA [83].

Intracranial aneurysms reveal an overexpression of several factors that align with extracellular matrix (ECM) remodeling. However, the presence of some of these factors is less anticipated. A case in point is OSF-2, a transcription factor that is typically associated with cells from the osteoblast lineage, hinting at its primary role in bone-related functions [84]. While OSF-2 has been implicated in the activation process of collagenase-3 (MMP-13) during bone development, its association with arterial conditions remains largely uncharted territory. As of now, no conclusive evidence ties it to arterial pathologies [85]. Venturing deeper into the cellular world, cathepsins emerge as notable entities. Specifically, Cathepsin B, a lysosomal cysteine protease, plays a pivotal role in intracellular protein breakdown. Its involvement has been noted in conditions like cancer and chronic inflammatory diseases, particularly those affecting airways and joints. On the other hand, Cathepsin D, an aspartyl endoproteinase, boasts a ubiquitous presence in lysosomes [86]. While cathepsins are primarily recognized for their protein-degrading prowess in lysosomes and phagosomes, their influence extends beyond this. They play a cardinal role in activating biologically active protein precursors within the prelysosomal compartments of specialized cells [87]. The association of cathepsins with tumor progression, especially in the context of invasion and metastasis, is hard to overlook. Their ability to dissolve the ECM grants them a pivotal role in such processes. Due to this, cathepsins have found utility as prognostic markers, offering insights into the potential metastatic tendencies of tumors [88]. Although cathepsins exhibit characteristics that are synonymous with tissue remodeling in vascular structures, their roles in arterial disorders have not been thoroughly explored. In the past, various studies have pointed towards their potential involvement in the pathogenesis of abdominal aortic aneurysms, prompting a closer examination of their significance in arterial conditions [89]. However, recent studies show that in the progression of abdominal aortic aneurysm (AAA) formation, there is a marked elevation in the expression and enzymatic activity of cathepsins, as substantiated by several studies [90,91]. Through the application of immunohistochemical techniques and a Western blot analysis, it was elucidated that cathepsin L is absent or negligibly expressed in healthy vascular tissues. However, in contrast, AAA tissues exhibited robustly positive staining for cathepsin L. Further, a quantitative assessment of the mRNA levels revealed that cathepsin L expression in AAA lesions was augmented by 22% relative to that in normal aortic tissues. Moreover, when comparing protein levels in AAA lesions to those in aorta tissues acquired from patients diagnosed with an artery occlusion, cathepsin H exhibited a substantial increase, registering at 330% higher in AAA lesions. This significant disparity underscores the potential role and importance of cathepsins in the pathogenesis and progression of an AAA [92].

### Inflammation Contribution

A defining characteristic of an intracranial aneurysm is the pervasive infiltration of inflammatory cells, which significantly contributes to its pathophysiology [64]. While various inflammatory cells find their way into the IA wall, macrophages stand out as the predominant infiltrators. Interestingly, both unruptured and ruptured IAs experience macrophage infiltration. However, ruptured IAs present a considerably higher degree of infiltration, highlighting a close link between vascular inflammation and the likelihood of IA rupture [93]. There is a pronounced correlation between the degree of leukocyte infiltration and degenerative changes in the IA wall. Such alterations include the notable loss of medial smooth muscle cells and the degradation of the extracellular matrix [94]. Studies utilizing animal models, such as mice and rats, have shed light on macrophages’ roles in IAs. Notably, the early stages of experimentally induced IAs in these displayed animals can stymie the degenerative changes observed in the IA wall. Moreover, mast cell degranulation has been linked to the enhanced expression and activation of matrix metalloproteinases (MMPs)-2 and -9, and inducible nitric oxide synthase (iNOS) in primary cultured SMCs. Given mast cells’ established roles in allergic inflammation, they seem to play integral roles in an IA’s inflammatory response [95]. An apparent link exists between the humoral immune reaction, driven by inflammatory cells, and an IA’s progression. Notably, antibodies like IgM and IgG predominantly line the luminal side of human IAs [96]. Tulamo et al.’s studies on human IA walls emphasized complement activation, especially in ruptured cases. They posited that this activation is correlated with an IA rupture and wall deterioration. As complement activation also triggers the release of pro-inflammatory cytokines, it might act as an ignition point for inflammatory cascades within the IA wall [97].

Endothelial cells are pivotal for maintaining vascular equilibrium. Significant morphological alterations in ECs within the IA wall have been documented, with noticeable gap formations at EC junctions. Furthermore, proteins that are integral for tight junctions, like occludin and zona occludens-1 (ZO-1), have been shown to be reduced in early stages of experimentally induced rat IAs [98]. Beyond mere morphological alterations, the functional impairment of the endothelium has garnered attention in an IA’s pathogenesis. Endothelial dysfunction is manifested through the activation of proinflammatory genes. For instance, certain chemokines and cell adhesion molecules, expressed in ECs within the IA wall, facilitate macrophage infiltration. Monocyte chemotactic protein-1 (MCP-1) stands out for its pivotal role in directing monocyte/macrophage traffic to areas impacted by various vascular conditions [99]. Early-stage rat IA formation showcases an upregulation in MCP-1 gene expression. Experiments with MCP-1-deficient mice revealed a marked decline in IA formation. Furthermore, treatments employing a dominant negative mutant of MCP-1 effectively hindered IA progression in rats [100]. To anchor monocytes/macrophages, vascular cell adhesion molecule-1 (VCAM-1) plays a significant role, fostering the firm adhesion of monocytes to ECs. Investigations into human IA samples have reported VCAM-1 expression, with gene expression profiling further underscoring its heightened presence in IAs [101]. The exact role that VCAM-1 plays in the development of an intracranial aneurysm is still under investigation. However, VCAM-1 is emerging as a pivotal molecule bridging endothelial dysfunction with the accumulation of macrophages. The transcriptional factor family, nuclear factor-kappa B (NF-κB), is integral to this process. It oversees the expression of numerous genes, including VCAM-1 and MCP-1, in response to inflammatory stimuli [102]. The signaling connection between prostaglandin E2 and prostaglandin E receptor 2 serves as a conduit between hemodynamic stress and IA genesis, primarily through the activation of NF-κB. PGE2 itself is birthed from arachidonic acid, thanks to the sequential actions of cyclooxygenase and PGE synthase. Of the many variants of these two enzymes, COX-2 and microsomal PGES1 (mPGES1)—which are usually upregulated in numerous inflammatory diseases—also exhibited increased expression in the walls of both rat and human IAs. This expression pattern coincided with that of EP2 [103]. Studies have shown that both an EP2 deficiency and COX-2 inhibition led to a significant curtailing of NF-κB activation and IA development. Intriguingly, COX-2 expression in the IA wall was suppressed by NF-κB inhibition, hinting at a positive feedback loop between COX-2 and NF-κB via EP2. Given that excessive shear stress can stimulate COX-2 and EP2 in cultured endothelial cells, it is plausible that this feedback mechanism, initiated by such stress, perpetuates chronic inflammation in the IA wall [104]. Research spearheaded by Jayaraman and colleagues has uncovered an elevation in TNF-α expression in human IA samples using both reverse transcription polymerase chain reaction (RT-PCR) and Western blotting methodologies [105]. The augmented presence of TNF-α was further corroborated in experimentally induced rat IAs. This increase was parallel to the heightened activity of the TNF-α converting enzyme (TACE), an enzyme that is instrumental in releasing TNF-α [106]. In the context of atherosclerosis, TLR-4 has been known to stimulate NF-κB activation in arterial walls. This expression was noted in the endothelial cell layers of both human and mouse IAs [107]. The concurrent expression patterns of TLR4 and NF-κB activation provide a solid ground for the hypothesis that the TNF-α/ΤLR4/NF-κB pathway could be instrumental in IA genesis. This, along with other observations, underscores the significance of TNF-α in IA pathogenesis [108]. Moreover, a study conducted by Wei et al. (2020) [109] showed that there is a connection between coronary artery ectasia (CAE) and inflammation, as well as between coronary artery aneurysm (CAA) and atherosclerosis, which are being intricately studied and are of interest in cardiovascular research. A notable characteristic of endothelial dysfunction is the diminished expression of eNOS. Rat IAs demonstrated a decline in eNOS expression when juxtaposed with the contralateral cerebral arterial wall devoid of IAs [110] (Figure 3).

IL-6, a pivotal proinflammatory cytokine, has been of interest in IA studies. Some researchers have shown an association between the IL-6 promoter polymorphism—572G>C—and IAs in Caucasians [111]. Moreover, the IL-6-572GG genotype was linked to a heightened IA risk in Chinese populations [112]. Both IL-12A and IL-12B seem to be implicated in IA susceptibility, either independently or jointly. Mounting evidence accentuates the importance of inflammatory processes in the onset and progression of IAs. Recent research has drawn connections between polymorphisms in the TP53 gene, inflammation, and inflammatory diseases [113]. Research has illuminated a significant association between IA and the rs6841581 polymorphism on chromosome 4q31.23, located just ahead of the endothelin receptor type A. Additionally, there is augmented evidence connecting IA to two other chromosomal regions: 12q22 and 20p12.1. These discoveries suggest that targeting the endothelin pathway could be pivotal for both the prevention and treatment of IAs [114].

A plethora of gene candidates implicated in the development of intracranial aneurysms are intertwined with the inflammatory cascade. This includes matrix metalloproteinases, transforming growth factor-beta (TGFβ) proteins, and endothelial nitric oxide synthase. Their presence underlines the functional significance of inflammation in the development of aneurysms. One primary mechanism involves the inflammatory response that becomes activated due to endothelial dysfunction and the degradation of the structural integrity of the cell wall [64]. There is a proposed model that outlines the process of aneurysm development initiated by inflammation. In this model, when inflammatory mediators are recruited to the cell wall, they disturb the internal elastic lamina, setting the stage for aneurysm formation. As the inflammation persists, the cellular wall undergoes more damage, leading to cell death and structural decay, and increasing the propensity for rupture. This model further gains strength when considering the role of nuclear factor-kappa B. Accumulating research underscores NF-κB’s importance in the genesis of cerebral aneurysms, particularly through its facilitation of the inflammatory process that recruits and activates macrophages [115,116]. The role of the immune response in the development and rupture risk of IAs is not just theoretical; empirical evidence also supports this claim. Advancements in RNA sequencing technologies have unearthed the involvement of certain pathways and components in IA pathogenesis. Specifically, studies spotlighted the lysosomal pathway and various immunoglobulins as key elements in the context of IAs. These findings strengthen the hypothesis that the immune system’s response is intricately woven into the tapestry of IA development and the subsequent risk of rupture [117].

Moreover, a key component that can contribute to the formation of an aneurysm through an inflammatory pathway is represented by the Renin–Angiotensin–Aldosterone System (RAAS).

A research study conducted by Cassis et al. (2009) [118] underscores a notable observation: ANG II-induced abdominal aortic aneurysms (AAAs) manifest through pathways that are not contingent upon blood pressure fluctuations. Furthermore, this investigation corroborates the notion that the exacerbation of atherosclerosis, when ANG II is introduced, transpires irrespective of the hypertensive effects attributed to ANG II.

It is scientifically fascinating to acknowledge that the infusion of ANG II can trigger two differentiated vascular malfunctions, both of which manifest without direct correlation to increased blood pressure metrics. Delving into the intricacies of ANG II’s broader implications, it is plausible to postulate that its primary impact might be anchored in inflammatory processes inherent to these variegated vascular dysfunctions.

In a broader therapeutic context, these findings illuminate a promising avenue. Medications that target and inhibit the renin–angiotensin system might present a dual advantage. According to Zhong et al. (2022) [119], for hypertensive patients with intracranial aneurysms, the administration of RAAS inhibitors notably reduced the risk of rupture, irrespective of their effect on blood pressure control.

## 5. Conclusions and Future Perspectives

The results from comprehensive genomic expression studies related to intracranial aneurysms have laid the groundwork for subsequent investigations. There is evidence suggesting that microRNAs play crucial roles in regulating vascular remodeling. This knowledge could be instrumental in pinpointing targets for potential treatments that might stabilize or decelerate the progression of an IA. It is also essential to identify the transcription factors that govern the transcriptional regulation of gene expression. The concentration of messenger RNA and microRNA in the blood samples of individuals with an IA might offer diagnostic solutions, acting as biomarkers. Such data could assist in discerning markers indicative of an impending rupture. Moreover, employing techniques like laser-capture microdissection to segregate distinct cell groups found in the IA wall would facilitate a comprehensive understanding of their roles in the pathobiology of IAs.

Intracranial aneurysms are perceived as multifaceted conditions typified by their onset, progression, and potential rupture. They are influenced by a myriad of genetic and external risk determinants that may affect one or multiple stages of the disease. While determining the impact of all recognized determinants on the likelihood of an individual developing an IA, or on the progression and potential rupture of an existing IA, it is imperative to also account for the interplay among these factors. This includes considering external risk determinants like tobacco usage. Undertaking such extensive analyses necessitates large cohorts with exhaustive data, demanding the collaboration of multiple research centers. The overarching objective of these molecular and genetic explorations is to harness the derived knowledge to construct advanced risk evaluation algorithms. These algorithms could anticipate an individual’s susceptibility to develop an IA, or monitor its potential progression or rupture, thereby proving to be invaluable in a clinical setting.

Another vital objective is to innovate therapeutic approaches based on insights into the mechanisms underlying the disease, with the aim to either preclude the onset or impede the progression of an IA. There is also a potential genetic commonality across diverse aneurysm types, as the same genetic variant on the 9p21 locus is associated with an increased risk for conditions such as myocardial infarction, abdominal aortic aneurysms (AAAs), and IAs. This suggests a predisposition towards a dysfunctional response to vascular injuries.

In recent times, the FIA consortium has propelled the study of the genetics of hereditary forms of IAs into the realm of whole-exome sequencing. Extensive familial histories, inclusive of American, Australian, and New Zealander populations affected by IAs, were amassed to scrutinize the entire coding sequences from fifty individuals, irrespective of whether they had an IA. Several pertinent genetic mutations were pinpointed, and there are ongoing efforts to further evaluate the affected genes [120].

There exists a comprehensive collection of studies in the scientific literature that associates genetic mutations and variations with the development and progression of intracranial aneurysms. The quality of these studies varies. It is likely that certain inherited genetic variations, with differing degrees of expressivity, contribute to the risk of IAs. The intricacies and full spectrum of interactions among multiple genes in relation to IAs still need to be explored in depth. Several genes have been identified as potential key players and deserve continued scrutiny. The rationale behind considering immune-altering agents for treatment might be based on the genetic underpinnings that drive IAs. Future research should focus on unbiased, comprehensive genomic analyses to uncover the roles of novel genes in the development and rupture of aneurysms. Such studies could offer crucial insights into potential biomarkers for this condition. As research progresses, there is a pressing need to create predictive models that integrate well-established genetic alterations. Given the advanced tools at our disposal in this era after the mapping of the human genome, comprehensive genomic studies involving large groups of patients diagnosed with IAs are essential to deepen our grasp on the genetic intricacies of this multifaceted disease.

In spite of significant strides in intracranial aneurysm research, the precise mechanisms through which blood flow induces inflammation remain elusive. It is unequivocal that changes in localized blood flow play pivotal roles in endothelial malfunction and adverse remodeling of the IA wall. The introduction of computational flow dynamics (CFD) in the early 2000s in IA research marked a significant advancement, offering a robust method for assessing flow characteristics within IAs and the associated mechanical properties of the aneurysm wall. Looking ahead, merging experimental approaches from molecular biology with engineering techniques holds promise to revolutionize our understanding of how flow dynamics influence biological alterations in the aneurysm wall. The specific mechanisms leading to IA rupture are still not entirely understood. Comparing ruptured and unruptured IAs has yielded insights into certain histopathological characteristics of those that rupture. In most unruptured IAs, inflammation and deterioration in the blood vessel wall undergo a natural healing process. Persistent inflammatory processes within the aneurysm wall, maintained by reinforcing feedback mechanisms, could provide insights into the molecular foundations of IA ruptures. In the coming years, imaging techniques such as MRI that can depict active inflammation within the aneurysm wall might become a pivotal diagnostic tool in identifying high-risk IAs that are more susceptible to rupture.

Moreover, screening should also be taken into account, since it plays an important role in preventing the rupturing of the aneurysm. Present guidelines do not recommend routine screening for intracranial aneurysms in the general public. However, they advise using CTA/MRA to screen for intracranial aneurysms in people who have at least two family members with intracranial aneurysms or subarachnoid hemorrhages. Additionally, screening is considered reasonable for patients with conditions like autosomal dominant polycystic kidney disease, fibromuscular dysplasia (FMD), aortic aneurysms, coarctation of the aorta, and microcephalic osteodysplastic primordial dwarfism, especially if they have a family history of these conditions [121].

## Figures and Tables

**Figure 1 brainsci-13-01454-f001:**
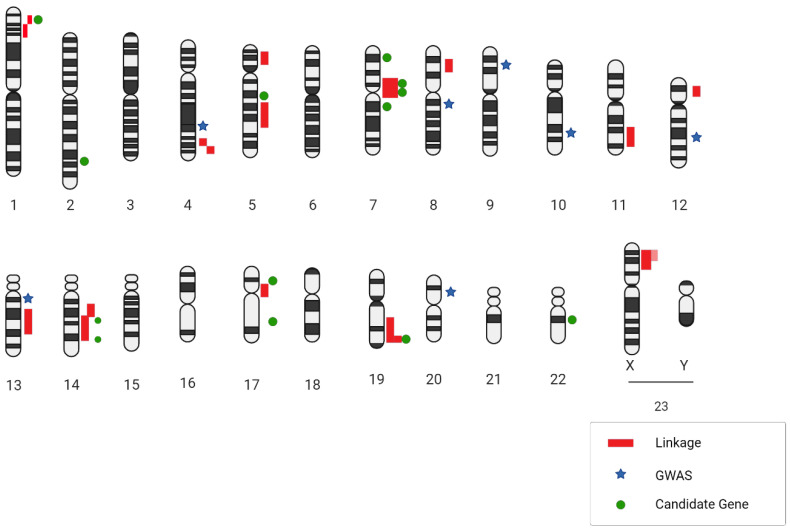
A genetic blueprint highlighting locations related to intracranial aneurysms is presented. Next to the chromosome diagrams, dark red lines point out regions identified through DNA linkage research. Light red lines suggest regions supported by family links to two specific spots. Blue star symbols mark the places where SNPs were identified in wide-ranging genetic studies, while green circular markers show where SNPs were pinpointed in targeted gene association studies.

**Figure 2 brainsci-13-01454-f002:**
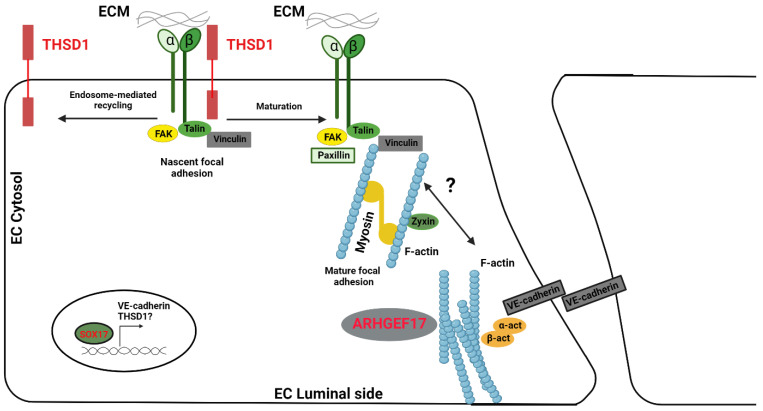
Diagram showing IA genes in vascular endothelial cells. Three key IA genes, THSD1, SOX17, and ARHGEF17, are emphasized in red. THSD1 interacts directly with the integrin complex via talin at emerging focal adhesions. As these focal adhesions mature, THSD1 relocates to a new emerging focal adhesion site through an endosome-based recycling mechanism. Without THSD1, there are impairments in focal adhesion, which is vital for the actin cytoskeleton and for influencing various subsequent pathways. SOX17 acts as a transcriptional regulator, affecting VE-cadherin expression. A decrease in VE-cadherin weakens cell-to-cell binding and raises permeability. ARHGEF17, a guanidine exchange factor, may play a role in the restructuring of the actin cytoskeleton.

**Figure 3 brainsci-13-01454-f003:**
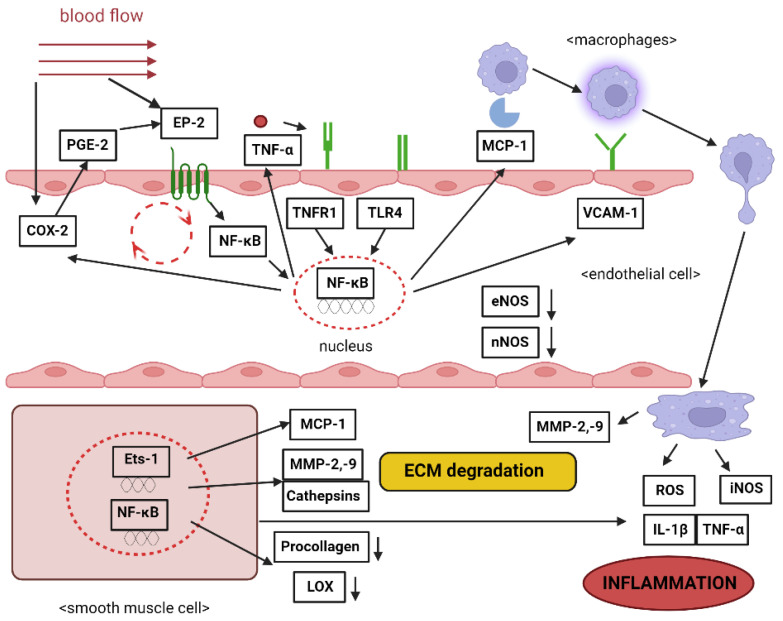
Inflammation pathways in the formation of intracranial aneurysms. COX-2 stands for cyclooxygenase-2; MCP-1 is known as monocyte chemotactic protein-1; PGE-2 refers to prostaglandin E2; MMP-2, -9 represent metalloproteinase types 2 and 9; EP-2 denotes prostaglandin receptor 2; ROS stands for reactive oxygen species; TNF-α is short for tumor necrosis factor-alpha; iNOS indicates inducible nitric oxide synthase; IL-1β refers to interleukin 1 beta; VCAM-1 represents vascular cell adhesion molecule-1; LOX is known as lysyl oxidase; NF-κB stands for nuclear factor kappa B; ECM is an abbreviation for extracellular matrix; TNFR1 is short for tumor necrosis factor receptor 1; TLR4 denotes Toll-like receptor 4; eNOS indicates endothelial nitric oxide synthase; nNOS stands for neuronal nitric oxide synthase.

**Table 1 brainsci-13-01454-t001:** Linkage analyses of DNA related to intracranial aneurysms.

Chromosomal Region	Study Design	LOD Score	Genetic Marker	Phenotype IDs and OMIM Locus
1p36.21-p36.13	Non-parametric	3.18	D1S2826-D1S234	609122; ANIB3
1p34.21-p36.13	Family, AD	4.2		609122; ANIB3
4q32.2	Non-parametric	2.5	Rs1458149	
4q32.3	Parametric	2.6		
5p15.2-p14.3	Family, AD	3.57	D5S1954	610213; ANIB4
5q22-q31	Affected sib pair	2.24	D5S471-D5S2010	
7q11	AR	2.34	D7S2421	105800; ANIB1
7q11	Affected sib pair	3.22	D7S2415-D7S657	105800; ANIB1
8p22	Family, AD	3.61	D8S552	614252; ANIB11
11q24-q25	Family	4.3	rs618176-rs1940033	612161; ANIB7
12p12.3	Parametric	3.1		
13q14.12-q21.1	Family	4.56	rs7983420-rs17054625	
14q23-q31	Family	3.0	rs235991-rs2373098	612162; ANIB8
14q22	Affected sib pair	2.31	D14S258-D14S74	
17cen	Non-parametric	3.0	D17S921-D17S1800	
19q13	Non-parametric	2.15	D19S198-D19S596	608542; ANIB2
19q13	Affected sib pair with covariates	5.70	D19S178-D19S545	608542; ANIB2
19q13.3	Affected only, parametric	4.10	D19S198-D19S902	608542; ANIB2
Xp22	Non-parametric	2.16	DXS987-DXS7593	300870; ANIB5
Xp22	Affected sib pair	2.08	DXS987	300870; ANIB5
Xp22.32-p22.2	Non-parametric	4.54	DXS6807-DXS1224	300870; ANIB5

AD = autosomal dominant; AR = autosomal recessive.

**Table 2 brainsci-13-01454-t002:** Key research findings on intracranial aneurysm genetics: loci, candidate genes, and associated functions.

Type of Genetic Analysis	Loci	Candidate Genes	Functions
GWAS	13q13.1	STARD 13	Movement of endothelial cells
GWAS	18q11.2	RBBP8	Cell cycle
GWAS	2q	PLCL1, BOLL	Similarity to phospholipase C, positioned after VEGFR2 in the signaling pathway
GWAS	4q31.23	EDNRA	Vasoconstriction
GWAS	8q21.3	SOX17	Endothelial sprouting
GWAS	9p21.3	CDKN2A/B	Smooth muscle proliferation
GWAS	10q24.32	CNNM2	Epithelial absorption of Mg^2+^
GWL	1p34.3-p36.13Xp22	PERLECAN	Promote endothelial cell growth and renewal, maintain the endothelial barrier function, and inhibit smooth muscle cell proliferation
GWL	7q11, 14q22, 5q22	ELASTIN	Elasticity of the parietal vessel
GWL	11q24, 14q23		
GWL	19q13, Xp22		

Abbreviations: GWL = genome wide linkage; GWAS = genome wide association studies.

## Data Availability

All data is available online on libraries such as PubMed.

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
