# Peer review of "Intracranial Aneurysms and Genetics: An Extensive Overview of Genomic Variations, Underlying Molecular Dynamics, Inflammatory Indicators, and Forward-Looking Insights"

_brainsci, 2023, doi:10.3390/brainsci13101454_

Round 1

Reviewer 1 Report

In effects, it is a relatively well written overview, but literature references are old (the most recent paper dates 2019 and most reported articles have been written before 2015). From 2019 to 2023, many important articles on intracranial aneurysms and genomics and other risk factors have been published, and none of them are reported and discussed. Nor are references to some important works of previous years reported (only as an example, some articles are attached at the bottom).

Moreover, the Authors do not report the methodology (study design, Search strategy and selection criteria) used. Furthermore, only some Genetic Syndromes with Increased Intracranial Aneurysm Incidence are discussed (Autosomal Dominant Polycystic Kidney Disease, Type IV Ehlers–Danlos Syndrome, Loeys Dietz Syndrome) neglecting many others (Marfan syndrome and other rarer ones and above all the familial forms or those associated with vascular anomalies in other districts).

It would be better if the authors want to update and discuss the references of recent literature, specify the research methodology, indicate the limitations of the study.

Suggested References:

1.     Clarke M. Systematic review of reviews of risk factors for intracranial aneurysms. Neuroradiology. 2008 Aug;50(8):653-64. doi: 10.1007/s00234-008-0411-9. Epub 2008 Jun 17. PMID: 18560819.

2.     Vlak MH, Algra A, Brandenburg R, Rinkel GJ. Prevalence of unruptured intracranial aneurysms, with emphasis on sex, age, comorbidity, country, and time period: a systematic review and meta-analysis. Lancet Neurol. 2011;10:626–636. doi: 10.1016/S1474-4422(11)70109-0.

3.     Hitchcock E, Gibson WT. A Review of the Genetics of Intracranial Berry Aneurysms and Implications for Genetic Counseling. J Genet Couns. 2017 Feb;26(1):21-31. doi: 10.1007/s10897-016-0029-8. Epub 2016 Oct 14. PMID: 27743245; PMCID: PMC5258806.

4.     Zhou S, Dion PA, Rouleau GA. Genetics of Intracranial Aneurysms. Stroke. 2018 Mar;49(3):780-787. doi: 10.1161/STROKEAHA.117.018152. Epub 2018 Feb 6. PMID: 29437983.

5.     Toth G, Cerejo R. Intracranial aneurysms: Review of current science and management. Vasc Med. 2018 Jun;23(3):276-288. doi: 10.1177/1358863X18754693. PMID: 29848228.

6.     Slot EMH, Rinkel GJE, Algra A, Ruigrok YM. Patient and aneurysm characteristics in familial intracranial aneurysms. A systematic review and meta-analysis. PLoS One. 2019 Apr 8;14(4):e0213372. doi: 10.1371/journal.pone.0213372. PMID: 30958821; PMCID: PMC6453525.

7.     Song J, Lim YC, Ko I, Kim JY, Kim DK. Prevalence of Intracranial Aneurysms in Patients With Systemic Vessel Aneurysms: A Nationwide Cohort Study. Stroke. 2020 Jan;51(1):115-120. doi: 10.1161/STROKEAHA.119.027285. Epub 2019 Nov 18. Erratum in: Stroke. 2020 Aug;51(8):e181. PMID: 31735136.

8.     Bakker MK, Ruigrok YM. Genetics of Intracranial Aneurysms. Stroke. 2021 Aug;52(9):3004-3012. doi: 10.1161/STROKEAHA.120.032621. Epub 2021 Aug 17. PMID: 34399582.

9.     Huguenard AL, Gupta VP, Braverman AC, Dacey RG. Genetic and heritable considerations in patients or families with both intracranial and extracranial aneurysms. J Neurosurg. 2021 Jan 1;134(6):1999-2006. doi: 10.3171/2020.8.JNS203234. PMID: 33386011

10.  Kumar M, Patel K, Chinnapparaj S, Sharma T, Aggarwal A, Singla N, Karthigeyan M, Singh A, Sahoo SK, Tripathi M, Takkar A, Gupta T, Pal A, Attri SV, Bansal YS, Ratho RK, Gupta SK, Khullar M, Vashishta RK, Mukherjee KK, Grover VK, Prasad R, Chatterjee A, Gowda H, Bhagat H. Dysregulated Genes and Signaling Pathways in the Formation and Rupture of Intracranial Aneurysm. Transl Stroke Res. 2023 Aug 30. doi: 10.1007/s12975-023-01178-w. Epub ahead of print. PMID: 37644376.

Author Response

Dear Reviewer,

Thank you for your positive feedback and kind suggestions,

We’ve addressed the mentioned issues, those modifications are highlighted

Moreover, the suggested references were added in the manuscript

Thank you for your significant contribution!

Reviewer 2 Report

Intracranial Aneurysms and Genetics: An Extensive Overview of Genomic Variations, Underlying Molecular Dynamics, Inflammatory Indicators, and Forward-Looking Insights.

This review is an attempt to condense the research and knowledge about intracranial aneurysms to provide insights for the future. While the review is extensive, the abstract does not say much. Please provide already some of your main findings.

Since there has already been a very nice review published on this topic, what do the authors now bring extra to light that is not mentioned in the previous review? Neuromolecular Med. 2019 Dec;21(4):325-343. doi: 10.1007/s12017-019-08537-7. This review also has a clear abstract on their findings, which is missing here.

In general it seems that the review could reduce its word count, because there is a lot of overlap statements and the use of too many words to say something simple.

The first sentence is already unclear and should be split in two: “About 80% of all non-traumatic subarachnoid hemorrhages (SAH) are caused by intracranial aneurysms (IAs), commonly referred to as saccular or berry aneurysms, between 2% and 5% of people have intracranial aneurysms.” Into: About 80% of all non-traumatic subarachnoid hemorrhages (SAH) are caused by intracranial aneurysms (IAs), commonly referred to as saccular or berry aneurysms. Between 2% and 5% of people have intracranial aneurysms.

Second sentence:” It's rare to find unruptured IAs in children, representing only 0.5% to 4.6% of all aneurysms. Their occurrence seems to increase with age [3].” What do the authors try to say here? It says something about the rupture rate of the IA in children, that the IA always rupture in children. And is says something about the incidence of IA in children, that it is rare in children and once people get older you find more IA. Is that really what the message is? I have the impression that the authors try to say that IA is rare in children and we hardly find IA that rupture in children.

Third sentence:” Women are more predisposed to aneurysms than men, with a 3:1 ratio in cases of unruptured IAs, while 70-75% of IAs appear individually, 25-30% occur as multiple aneurysms [5],[6].” Does all this apply only to women? Or is part of the info about women and the rest of the sentence about IA in general in men and women? Please split sentence into two if the latter is meant for all IA.

These type of sentences occur often. Please have a native English person look at the manuscript to refine the text and make it more specific and condense.

What is interesting is that abdominal aortic aneurysms, which are also highly inflammatory, are predominantly present in men. So IA seems a female inflammatory vascular disease and AAA a male inflammatory vascular disease. This suggests other upstream pathways to induce the inflammation, or a process that would trigger the inflammatory response. This is not mentioned anywhere. (In Marfan syndrome men also have more severe thoracic aorta aneurysm disease progression; Genet Med. 2021 Jul;23(7):1296-1304. doi: 10.1038/s41436-021-01132-x.)

Many genetic syndromes are mentioned, but most of them are not discussed further on. On the other hand some new syndromes are discussed that were not mentioned here? Please mention all syndromes at first and then highlight why only a few are later discussed.  “Although several inheritable conditions, including autosomal dominant polycystic kidney disease, neurofibromatosis type I, Marfan syndrome, multiple endocrine neoplasia type I, pseudoxanthoma elasticum, hereditary hemorrhagic telangiectasia, and Ehlers Danlos syndrome types II and IV, have links to the formation of IAs, they represent less than 1% of all IAs in the general population.”

Microcephalic/Majewski’s Osteodysplastic Primordial Dwarfism, Type II (MOPD2) is not underlined. And this paragraph should be immediately after PKD and before type IV EDS, since the functionality is similar. This also goes for EDS and LDS, which belong together and should thus be discussed one after the other.

Just to provide some more insight: in PKD and MOPD2 the cilia are disrupted. Cilia are hairs on cells to sense flow in cells that deal with motion, such as endothelial cells in the vasculature and epithelial cells in the kidney. If these cells cannot sense flow they adapt their behavior, such as proliferate to form cysts in the kidney. A similar feature can be envisioned for the effect on the vasculature. Disturbed function can lead to activation of endothelial cells and attraction of inflammatory cells. Atherosclerosis. 2018 Aug;275:196-204. doi: 10.1016/j.atherosclerosis.2018.06.818. In short, this means that the mechanosensing pathways are disturbed in the vasculature, which are essential for proper function and suppression of inflammation. In that light also the syndromes related to connective tissue should be seen. Dysregulated extracellular matrix such as in EDS, LDS and Marfan means that the local surroundings are different and the cells feel this, thus their mechanosensing is different because the extracellular matrix is more or maybe less stiff, which translates into other signals in the cells. (Obviously, in LDS there is also disturbed TGFbeta signaling influencing other pathways then the TGF-dependent ECM production. ) Science. 2014 May 2;344(6183):477-9. doi: 10.1126/science.1253026. / Cardiovasc Drugs Ther. 2021 Dec;35(6):1233-1252. doi: 10.1007/s10557-020-07116-4. The bottom line is that in all these diseases the mechanosensing is disturbed and cues from outside of the cell are translated wrong into the cell, which disturbed cell behavior. Thus there is a clear red line and I miss the extrapolation on that to actually create a new sense of understanding of the genetics.  

On the role of angiotensin/RAAS and aneurysms, it is of interest that even independent of hypertension the RAAS system can induce aneurysms in mice due to promotion of inflammation (vasculitis due to massive bone marrow release of inflammatory cells). Am J Physiol Heart Circ Physiol. 2009 May;296(5):H1660-5. doi: 10.1152/ajpheart.00028.2009. So prolonged activation of the RAAS system also promotes a vascular inflammatory response, which seems relevant in the inflammatory IA.

“The EDNRA gene encodes a receptor that is activated..”, but one paragraph later the abbreviation is mentioned: “The SNP denoted as rs6841581A.G, located on chromosome 4q31.23, which codes for the endothelin receptor type A (often abbreviated as EDNRA)…”

Table 2 . Perlecan is not only involved in EC function; the Gene Cards website states: involved in maintaining the endothelial barrier function and inhibitor of smooth muscle cell proliferation and is thus thought to help maintain vascular homeostasis. Please mention function for both cell types per gene to judge if there is a real endothelial mark here or not. That would be interesting, since the thoracic aneurysm genetic diseases have more an ECM/smooth muscle cell mark. And perhaps here there is an ECM/endothelial cell mark.

THSD1 is also an ECM protein (and produced by SMCs), thus it may also regulate the focal adhesions via changing the ECM. That seems rather similar as perlecan function.

“However, a recent revelation points towards their potential involvement in the pathogenesis of ab- 460 dominal aortic aneurysms, prompting a closer examination of their significance in arterial  conditions [73]” The manuscript that is sited is from 1997 and thus not recent. It has indeed been known for a long time that cathepsins may play a role in aneurysmal disease, there are many papers on this topic.

“medial Smooth Muscle Cells and the degradation of the Extracellular Matrix. [75].” Why in capitals?

The inflammation part could be written for atherosclerosis, since the same genes pop up there. Yet, women are normally protected somewhat from atherosclerosis, so what makes the inflammation different in atherosclerosis or in IA? Do women just have atherosclerosis at other sites than men? Or is the trigger for IA-related inflammation unrelated to atherosclerosis, but it triggers a similar inflammatory response? I think there should be more focus on research of this male/female discrepancy with other vascular diseases.

If I read this review my main conclusions for IA are; 1) that women are far more at risk than men is very intriguing, since in other aneurysm diseases and atherosclerosis women are protected. And 2) that genes associated with IA have more an ECM/endothelial cell signature, as compared to other genetic aneurysm diseases which have more an ECM/smooth muscle cell profile. Yet, both point at defects in mechanosensing of the cells.

 The language is fine, but some sentences are too long and it is written as a novel. However, in view of how much time we have to keep up with the literature I encourage the authors to condense their manuscript. The article is too long and could be reduced in wordcount by 25% and be more factual.

Author Response

Dear Reviewer,

Thank you for your positive feedback and kind suggestions,

We’ve addressed the mentioned issues, those modifications are highlighted

Additionally, the paragraphs mentioned were improved according to your recommendations

Thank you for your significant contribution!

Round 2

Reviewer 2 Report

I have nothing further